# Radiomics Signature of Aging Myocardium in Cardiac Photon-Counting Computed Tomography

**DOI:** 10.3390/diagnostics15141796

**Published:** 2025-07-16

**Authors:** Alexander Hertel, Mustafa Kuru, Johann S. Rink, Florian Haag, Abhinay Vellala, Theano Papavassiliu, Matthias F. Froelich, Stefan O. Schoenberg, Isabelle Ayx

**Affiliations:** 1Department of Radiology and Nuclear Medicine, University Medical Center Mannheim, Heidelberg University, Theodor-Kutzer-Ufer 1-3, 68167 Mannheim, Germany; mustafa.kuru@umm.de (M.K.); johann.rink@medma.uni-heidelberg.de (J.S.R.); florian.haag@medma.uni-heidelberg.de (F.H.); abhinay.vellala@medma.uni-heidelberg.de (A.V.); matthias.froelich@medma.uni-heidelberg.de (M.F.F.); stefan.schoenberg@medma.uni-heidelberg.de (S.O.S.); isabelle.ayx@medma.uni-heidelberg.de (I.A.); 2First Department of Medicine-Cardiology, University Medical Center Mannheim, Heidelberg University, Theodor-Kutzer-Ufer 1-3, 68167 Mannheim, Germany; theano.papavassiliu@umm.de

**Keywords:** myocardial aging, cardiac CT, photon-counting computed tomography, radiomics

## Abstract

**Background**: Cardiovascular diseases are the leading cause of global mortality, with 80% of coronary heart disease in patients over 65. Understanding aging cardiovascular structures is crucial. Photon-counting computed tomography (PCCT) offers improved spatial and temporal resolution and better signal-to-noise ratio, enabling texture analysis in clinical routines. Detecting structural changes in aging left-ventricular myocardium may help predict cardiovascular risk. **Methods**: In this retrospective, single-center, IRB-approved study, 90 patients underwent ECG-gated contrast-enhanced cardiac CT using dual-source PCCT (NAEOTOM Alpha, Siemens). Patients were divided into two age groups (50–60 years and 70–80 years). The left ventricular myocardium was segmented semi-automatically, and radiomics features were extracted using pyradiomics to compare myocardial texture features. Epicardial adipose tissue (EAT) density, thickness, and other clinical parameters were recorded. Statistical analysis was conducted with R and a Python-based random forest classifier. **Results**: The study assessed 90 patients (50–60 years, *n* = 54, and 70–80 years, *n* = 36) with a mean age of 63.6 years. No significant differences were found in mean Agatston score, gender distribution, or conditions like hypertension, diabetes, hypercholesterolemia, or nicotine abuse. EAT measurements showed no significant differences. The Random Forest Classifier achieved a training accuracy of 0.95 and a test accuracy of 0.74 for age group differentiation. Wavelet-HLH_glszm_GrayLevelNonUniformity was a key differentiator. **Conclusions**: Radiomics texture features of the left ventricular myocardium outperformed conventional parameters like EAT density and thickness in differentiating age groups, offering a potential imaging biomarker for myocardial aging. Radiomics analysis of left ventricular myocardium offers a unique opportunity to visualize changes in myocardial texture during aging and could serve as a cardiac risk predictor.

## 1. Introduction

Until today, cardiovascular diseases remain the leading cause of global mortality in developed countries [1]. Apart from obvious and modifiable cardiovascular risk factors such as nicotine abuse, hypertension, or hypercholesterolemia [2], aging is a severe risk factor for developing cardiovascular disease. Four out of five people who are suffering from coronary heart disease are older than 65 years [3]. Especially due to the increase in the elderly population, a better understanding of the changes of cardiovascular structure during aging is relevant in addressing this risk factor in the future.

Coronary computed tomography angiography (CCTA) offers high diagnostic accuracy in detecting significant coronary artery stenosis exceeding 50% and notably excels in excluding obstructive coronary artery disease with a nearly perfect negative predictive value of almost 100% [4,5]. Consequently, the recognized incremental value of CCTA in detecting and ruling out coronary artery stenosis has prompted revisions in cardiac CT guidelines in the last few years [6]. However, despite the far-reaching advances in cardiac CT imaging, the analysis of the left ventricular myocardium remains the domain of magnetic resonance (MR) imaging [7]. To address this limitation, recent studies have focused on assessing myocardial changes in cardiac CT. In this context, additional late-iodine acquisitions have been used to delve deeper into myocardial analysis [8]. Nevertheless, the need for an analysis of the left-ventricular myocardium in the already gained CCTA images is unbroken.

In recent years, texture analysis has evolved as a promising tool in times of big data. Radiomics analysis is a technique that extracts numerous features from a region of interest to measure hundreds of parameters from radiological images [9]. Through radiomics analysis, texture changes not visible to the human eye can be detected and may serve as additional tools in identifying patients at risk. Radiomics analysis has been widely used in oncologic settings [10,11], but it is increasingly integrated in cardiac imaging as well. Multiple studies have demonstrated the ability to detect myocardial fibrosis through texture analysis on CCTA [12,13,14,15,16]. Additionally, the influence of various cardiovascular risk factors on left-ventricular myocardium on MRI has been demonstrated in the past [17]. Nevertheless, the widespread adoption of texture analysis in clinical practice has been impeded by issues of reproducibility and feature stability, with various factors such as reconstruction algorithms, contrast phases, and scanner types influencing radiomics features [18,19,20,21]. In this context, the integration of artificial intelligence (AI) into clinical routine is of increasing interest, summarizing machine learning, artificial neural networks, and deep learning techniques to enhance diagnostic and treatment processes [22].

The integration of photon-counting detector CT (PCCT) technology may overcome these challenges of radiomics analysis and introduce novel opportunities in cardiac CT imaging. Distinguished by its ability to detect individual photons and directly convert them into electric signals without intermediate steps of converting the photons into visible light, PCCT offers superior spatial and temporal resolution, reduced beam-hardening artifacts, and enhanced signal-to-noise ratio compared to conventional energy-integrating CT (EICT) [23,24,25]. This technique offers the possibility for a more stable radiomics analysis and addresses the recent limitations of reduced reproducibility and feature stability [26].

A recently even more recognized tissue in the context of cardiovascular disease development is the epicardial adipose tissue (EAT). The EAT is the anatomical fat reservoir between the visceral layer of the pericardium and the surface of the left myocardium [27]. During recent years, various studies have focused on the connection between EAT and the pathogenesis of CVD. In this context, the density and volume of the EAT have been of particular interest [28], as various studies were able to detect an association between EAT density and volume with an elevated Agatston score, an increased risk of CAD, and the prediction of obstructive CAD and high-risk plaque features [29].

As longevity is of rising interest in the medical community and the focus is led from an aging society to a longevity society, there is an urgent need to better understand the parameters of myocardial aging [30]. Potential myocardial changes during aging might become an identifier for unhealthy aging and a potential target for long-lasting longevity therapies. Hence, this study aims to explore potential age-dependent texture changes of the left-ventricular myocardium as well as potential differences in EAT density and thickness to identify novel cardiac biomarkers.

## 2. Material and Methods

### 2.1. Patients Collective

In this retrospective single-center study, patients with clinically indicated electrocardiography (ECG)-gated contrast-enhanced cardiac CT according to ESC guidelines were enrolled between August 2022 and December 2023 [31]. In total, 90 patients (42 female, mean age 65.6 (range: 51–79 years)), were included in this study. Out of these 90 patients, 54 patients were between 50 and 60 years old, whereas the other 36 patients were between 70 and 80 years old. The age categories were defined using 10-year intervals to avoid borderline cases and ensure clear and unambiguous classification into distinct age groups, thereby facilitating more accurate analysis of aging-related factors. For each patient, various clinical cardiovascular risk factors as well as the Agatston score were noted. Table 1 offers an overview of the detailed patients’ criteria. This retrospective study was approved by an institutional review board and local ethics committee (ID 2021-659, date 26 October 2021). All investigations were conducted according to the Declaration of Helsinki. Figure 1 shows the study workflow.

### 2.2. Cardiac CT Imaging

All patients were scanned using the first-generation whole-body dual-source PCCT (NAEOTOM Alpha, Siemens Healthcare GmbH, Forchheim, Germany). The tube voltage was set at 140 kV. Gantry rotation time was 0.25 s, and automatic dose modulation was used. The primary performed unenhanced scan was used to estimate the Agatston score and exclude patients with an Agatston score above 100, as an Agatston score of 100 has been set in literature as a threshold, as most coronary events have been associated with a calcium score of greater than 100 [16,32] and the influence of coronary artery calcifications on left-ventricular myocardium was already proven and should not influence the underlying study. In all patients, ß-blocker (5–10 mg Metoprolol, Recordati Pharma GmbH, Ulm, Germany) was applied to lower the heart rate below 65 beats per minute. Additionally, sublingual nitroglycerin (0.5 mL) was applied directly before scanning. An iodinated contrast medium (70–80 mL Imeron 400, Bracco Imaging Deutschland GmbH, Konstanz, Germany) was injected, followed by a saline chaser (20 mL, NaCl 0.9%) with a weight-based flow rate (5–6 mL/s) via an antecubital vein. Bolus tracking to trigger the start of CCTA was achieved by placing a region of interest (ROI) in the ascending thoracic aorta (threshold 140 HU at 90 kV).

### 2.3. Cardiac CT Image Analysis

The non-contrast enhanced scan was used to estimate the Agatston score using axial non-enhanced slices with 3 mm slice thickness, Qr36 kernel, and dedicated software (syngo.via (version VB80), Siemens Healthcare GmbH, Forchheim, Germany). Axial images of contrast-enhanced CCTA were reconstructed using a soft vascular kernel (Bv40), matrix size of 512 × 512, slice thickness of 0.6 mm, and increment of 0.4 mm. These images were anonymized, exported, and stored in digital imaging and communication in medicine (DICOM) file format. These data were converted to Neuroimaging Informatics Technology Initiative (NIFTI) file format for usability with an open-source segmentation tool (3D Slicer, Version 4.11, The 3D Slicer Community, https://www.slicer.org, 2020). In each patient, the whole left-ventricular myocardium was segmented semi-automatically by a radiologist with 4 years of experience in cardiovascular imaging. To ensure the quality and consistency of the segmentations, all results were reviewed in a structured manner by a second radiologist with over a decade of experience in cardiovascular imaging and more than eight years of expertise in image segmentation. EAT density was measured with standardized regions of interest (ROI) with an area of 0.4 cm^2^. Additionally, the thickness of the epicardial fat tissue was measured at 3 previously defined sites of the right ventricle. Figure 2 shows an example segmentation of a patient from the age group 70–80 years (A) as well as an example measurement of the density (B) and thickness of the epicardial adipose tissue (C).

### 2.4. Radiomics Feature Extraction

Radiomics features were extracted from all segmentations using the PyRadiomics framework (version 3.0.1). For each patient, first-order statistics as well as texture features derived from matrices such as the gray level co-occurrence matrix (GLCM), gray level dependence matrix (GLDM), gray level size zone matrix (GLSZM), gray level run length matrix (GLRLM), and neighboring gray tone difference matrix (NGTDM) were computed. Shape-based descriptors were intentionally excluded, as the focus of this analysis was on age-related variations in tissue texture. Feature extraction was conducted using PyRadiomics’ default parameters, following the guidelines of the Image Biomarker Standardisation Initiative (IBSI) [33]. The preprocessing included resampling to an isotropic voxel size of 1 mm^3^, discretization of gray levels using a fixed bin width of 32, and application of a mask dilation radius of one voxel.

### 2.5. Statistical Analysis

The descriptive statistics of the patients were analyzed using R Studio (Version 2024.04.1+748, RStudio Team (2020), RStudio: Integrated Development for R. RStudio, PBC, Boston, MA, USA, URL http://www.rstudio.com/), utilizing the “tableone” package) [34]. The default hypothesis tests are chi-squared tests for categorical variables with continuity correction and ANOVA tests for continuous variables under the equal variance assumption. For non-normal variables and small cell counts, Kruskal–Wallis tests were used for non-normal continuous variables, and Fisher’s exact tests for categorical variables were specified in the exact argument. The Kruskal–Wallis test is equivalent to the Wilcoxon test for two-group comparisons. For the statistical analysis of the continuous parameters of the Agatston score and the measurements of epicardial adipose tissue, Shapiro–Wilk tests and distribution analyses were first used to determine whether these are symmetrically or asymmetrically distributed. Subsequently, a Mann–Whitney U test or *t*-test was carried out to determine whether there were significant differences between the parameters and the age groups.

In addition, a Python-based random forest (RF) classifier was applied, and a systematic hyperparameter tuning approach using grid search cross-validation was utilized to optimize its performance. The dataset was randomly split into 75% training and 25% test data, using stratified sampling to maintain the original distribution of the two age groups (60% and 40%, respectively) in both subsets. Hyperparameter optimization for the random forest classifier was performed via internal cross-validation within the training set. A confusion matrix was then created for the test data to provide a more detailed breakdown of sensitivity and specificity (Figure 3), and a feature importance analysis was performed (Figure 4). Additionally, a receiver operating characteristic (ROC) analysis was performed on the test dataset to evaluate the discriminative performance of the classifier, and the area under the curve (AUC) was calculated.

## 3. Results

### 3.1. Patient Population

Patient characteristics of the 90 included patients with 54 patients in age group 1 (50–60 years) and 36 patients in age group 2 (70–80 years) are outlined in Table 1. The overall mean age was 63.6 years, with a range of 51–79 years. The average age in age group 1 was 56.5 years, and in group 2, it was 74.2 years and differed significantly (*p* < 0.001). The average Agatston score was 28.9 overall, 29.1 in age group 1, and 28.5 in age group 2 (*p* = 0.921). There were no significant differences concerning gender distribution (*p* = 0.052) or relevant pre-existing conditions such as arterial hypertension (*p* = 0.764), hypercholesterolemia (*p* = 0.537), diabetes mellitus (*p* = 0.068), or nicotine abuse (*p* = 0.506).

### 3.2. EAT Analysis

There was no significant difference between the measured density values of the epicardial adipose tissue, neither in the mean (*p* = 0.914), in the standard deviation (*p* = 0.454), the measured minimum density (*p* = 0.5614), nor the maximum density (*p* = 0.227). There were also no significant differences in the mean average of the measured diameters of the epicardial adipose tissue at position 1 (*p* = 0.231), position 2 (*p* = 0.613), or position 3 (*p* = 0.664). Table 2 shows the results of these measurements.

### 3.3. Radiomics Analysis

With the random forest classifier used, we were able to achieve a train data accuracy of 0.95 and a test data accuracy of 0.74. In the test data, precision for age group 1 was 0.74 and for age group 2 was 0.75. Test data recall was 0.88 for age group 1 and 0.55 for age group 2. F1-Score was 0.8 for age group 1 and 0.63 for age group 2. Table 3 shows the results of the RF classifier in detail.

The feature importance analysis (Figure 4) identified several radiomics features that were important for differentiating between the two age groups studied. In particular, wavelet-HLH_glszm_GrayLevelNonUniformity should be mentioned here, which is a measure of the variability of the measured gray level intensity values, and a lower value indicates a higher degree of homogeneity. Mean values of Wavelet-HLH_glzm_GrayLevelNonUniformity were lower in the older population (1175.317), indicating a more heterogeneous texture of the left-ventricular myocardium in the younger population (1431.954). Regarding the minimum and maximum value, no patient in group 2 exceeded the value of 1921, and no patient in group 1 fell below 729.

To further assess the performance of the classifier, a receiver operating characteristic (ROC) analysis was performed on the test dataset. The resulting area under the curve (AUC) was 0.71, indicating moderate discriminatory ability in distinguishing between the two age groups based on the extracted radiomics features. The corresponding ROC curve is presented in Figure 5.

## 4. Discussion

In this study, we were able to differentiate between the two age groups (50–60 years vs. 70–80 years) using a random forest classifier based on the extracted radiomics parameters of the segmented left ventricular myocardium in PCCT scans, achieving a training data accuracy of 0.95 and a test data accuracy of 0.74. Age group 1 can be predicted more accurately than age group 2 in the test data. This is likely explained by the fact that the myocardium of the younger age group appears more heterogeneous in the PCCT scans compared to the older age group. Accordingly, in the feature importance analysis, wavelet-HLH_glszm_GrayLevelNonUniformity emerged as a particularly important parameter for differentiating the two age groups, representing a measure of the homogeneity of the extracted tissue. Even though a clear threshold could not be defined, particularly high or low values (below 729 or above 1921) could indicate the aging progress. On the other hand, conventional analysis of EAT, linked to cardiovascular diseases in general, did not offer the possibility of differentiation between both groups. While the feature importance analysis identified Wavelet-HLH_glszm_GrayLevelNonUniformity and other parameters as highly relevant for classification, we further investigated whether limiting the model to the top-ranked features would improve performance. Therefore, we trained a separate random forest classifier using only the top 50 features from the importance ranking. This model achieved a test accuracy of 0.70, which was notably lower than the original model that used the full feature set (accuracy: 0.74).

This finding indicates that although certain features contribute more strongly to the classification, the predictive performance of the model relies on the full multidimensional radiomic signature, and excluding lower-ranked features results in a loss of discriminative information. It also supports the idea that myocardial ageing, as captured in radiomic patterns, may be a diffuse and complex process that cannot be sufficiently described by a limited subset of features alone [33,35].

These results emphasize the importance of holistic feature modeling when working with high-dimensional radiomics data and caution against premature reduction to only the most prominent features. Future studies with larger datasets may help further refine feature selection strategies and investigate whether reduced models can reach similar or improved generalizability.

The idea of identifying a radiomics signature of cardiovascular risk factors in cardiac imaging has already been studied in the past. Cetin et al. analyzed the left and right ventricles in cardiac MRI of 5.065 UK Biobank participants [17]. Depending on various cardiovascular risk factors such as hypertension, diabetes, high cholesterol, and current or previous nicotine abuse, they were able to identify risk-specific radiomics signatures by comparing the radiomics analysis of short-axis images of the left and right ventricles at end-diastole and end-systole with a randomly matched healthy control group. For each group, the most discriminatory radiomics feature was identified and interpreted in a clinical context. The surface area-to-volume ratio was smaller in hypertensive individuals and previous smokers in comparison to the control groups, potentially reflecting a pattern of global concentric hypertrophy in affected patients. Diabetes patients could be discriminated by the median intensity of the myocardium at end-systole revealing a potential global alteration in the myocardial tissue. As a prominent feature, gray-level non-uniformity has been identified as a discriminating feature in current smokers, outlining an increase of heterogeneity in gray-level intensities in smokers—as it has been previously reported in patients with hypertrophic cardiomyopathy [36], suggesting that one feature might not be important for only one condition. Compared to our study, this study did not consider aging as an underlying risk factor. However, in line with our results, gray-level non-uniformity was a prominent feature in both studies in differentiating between various risk factors. In our study, this feature could differentiate aging myocardium, while in the study by Cetin et al. [17], it was used to discriminate current smokers. In both cases, the heterogeneity of the myocardium was the relevant parameter outlining a potential diffuse fibrotic change of myocardial texture through these risk factors.

A study from our institution already investigated the association between coronary artery calcifications and the texture of the left ventricular myocardium in the past. CCTA offers the unique possibility to estimate the degree of coronary artery calcifications and contemporaneous evaluation of the left ventricular myocardium. Patients were divided into three groups depending on the amount of coronary artery calcifications, and the left-ventricular myocardium was segmented semi-automatically. Random forest feature selection identified a set of four different myocardial features, which were associated with the degree of coronary artery calcification. It was even evident that the amount of coronary artery calcifications tended to correlate with the change of left-ventricular myocardial texture, indicating a potential severity-associated effect [16]. To exclude this bias from our study, we matched the Agatston scores of the two populations. The difference in myocardial texture of our study is hence not due to conventional cardiovascular risk factors, as far as they could be collected. Dedicated functional markers, such as diastolic and systolic function, or structural myocardial changes, such as concentric remodeling, have not been investigated and could still correlate with the myocardial texture.

Sangaralingham et al. have already addressed the problem of aging coronary vasculature in cardiac micro-computed tomography in Fischer rats in 2012. Young and aged Fischer rats underwent cardiac micro-CT in their study, accompanied by echocardiography, blood pressure measurement, and fibrosis analysis. They outlined an increase in LV mass and epicardial vessel volume but a reduction of total and intramyocardial vessel volume with age. Additionally, there was significant LV fibrosis (*p* < 0.05) and mild LV dysfunction in aged hearts in comparison to young hearts [37]. However, compared to our study, there was no detailed texture analysis and no reference to the human heart. In line with these results, our study was able to outline heterogeneity as a discriminating tool in aging myocardium. This might be due to relevant myocardial fibrosis in aging persons and might serve as an imaging biomarker correlating to the amount of myocardial fibrosis in the future.

An underlying study about the echocardiographic assessment of age-associated changes in Fischer rats was published in 2003 by Boluyt et al. [38]. In their study, aged Fischer rats presented a significant decrement in the resting systolic function of the LV and declines in LV ejection fractions, fractional shortening, and velocity of circumferential fiber shortening in comparison to young rats. Additionally, there was an increase in isovolumic relaxation time as a parameter for diastolic function in aged rats. They found that the modest changes in systolic and diastolic LV function during aging in rats reduce the capacity of the heart to respond to hemodynamic challenges.

Nevertheless, this study is hampered by various limitations, primarily due to the retrospective and single-center nature of this preliminary study, with a relatively modest size of the study cohort due to the recent implementation of the PCCT scanner. The limited global availability of the recently implemented PCCT technology reflects broader challenges in healthcare infrastructure worldwide. Due to its current restricted accessibility, the integration of PCCT findings into routine clinical practice remains limited. Although initial progress has been made through the development of more cost-effective alternatives, widespread implementation is still a long-term goal [39]. As a result, the benefits and insights gained from PCCT are, for the time being, largely confined to specialized centers.

One relevant limitation in this context is the challenge of reproducibility associated with radiomics analysis. However, a phantom study recently outlined elevated stability of radiomics features using PCCT [26]. Additionally, clinical data were collected using questionnaires and medical records; hence, some clinical data, such as detailed medication for example, might not be fully captured. Dedicated inflammatory biomarkers, preventive medical therapy, and the genetic background have not been registered and could influence the myocardial texture. Additionally, there was a mismatch in the prevalence of diabetes between both groups, which could induce fibrotic myocardial changes and cardiomyocyte hypertrophy. Future studies should investigate the texture changes of myocardial aging in a prospective multicenter approach involving a larger study population to address the current limitations. Furthermore, analysis of functional or structural myocardial changes using established tools like strain or indexed systolic function should validate these preliminary findings of aging myocardium and might be useful as predictors of clinical outcomes.

## 5. Conclusions

In conclusion, this study demonstrates the feasibility of differentiating left-ventricular myocardium according to different age groups based on differences in heterogeneities of the underlying texture. The parameter wavelet-HLH_glszm_GrayLevelNonUniformity might function as a potential imaging biomarker of myocardial aging in the future.

### Key Points

Radiomics texture features of the left ventricular myocardium in photon counting CT outperform traditional metrics like epicardial adipose tissue density and thickness in distinguishing age groups.

A random forest classifier achieved 0.95 training and 0.74 test accuracy in age group classification, highlighting wavelet-HLH_glszm_GrayLevelNonUniformity as an important feature.

Radiomics of myocardial texture can visualize aging changes and improve cardiovascular risk profiling for personalized treatment.

## Figures and Tables

**Figure 1 diagnostics-15-01796-f001:**
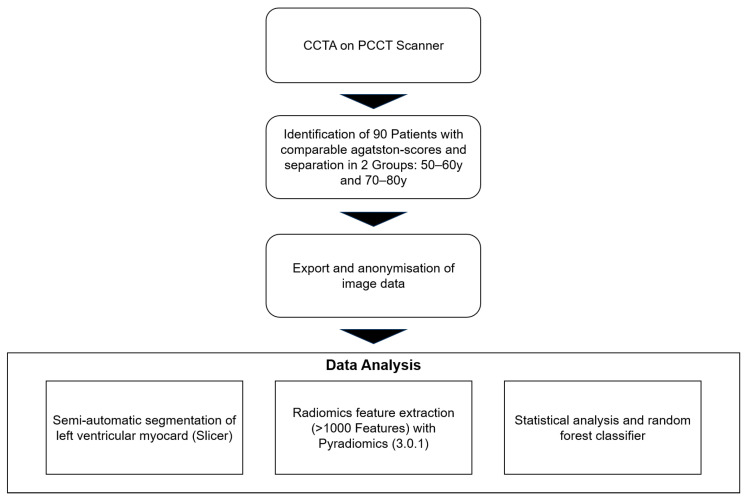
Study design. Abbreviations: CCTA: coronary computed tomography angiography, PCCT: photon-counting computed tomography.

**Figure 2 diagnostics-15-01796-f002:**
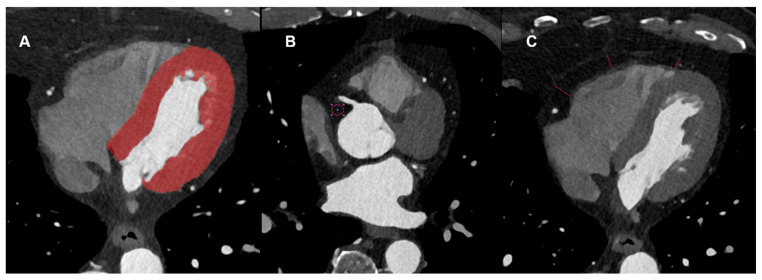
Example Segmentation of the left ventricular myocardium (**A**) as well as measurements of the HU density (**B**) and epicardial adipose tissue diameter (**C**).

**Figure 3 diagnostics-15-01796-f003:**
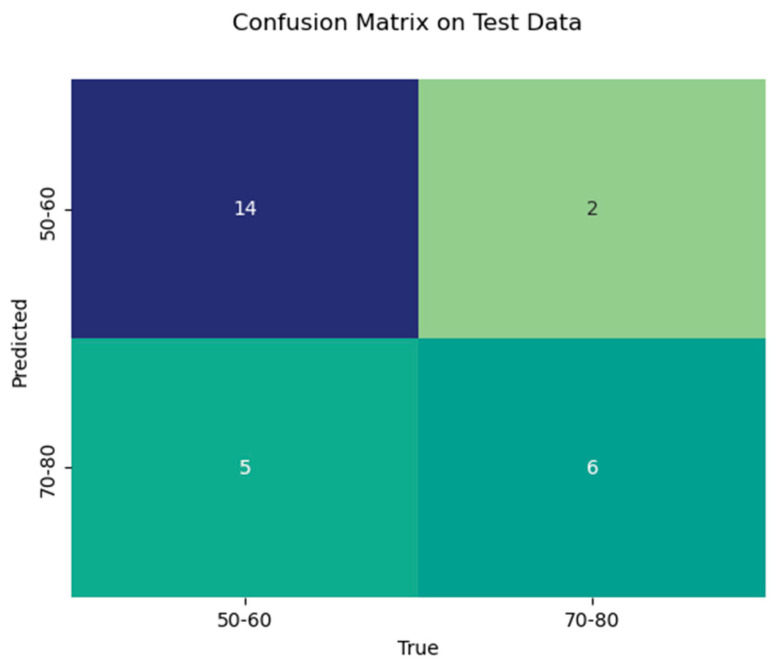
Confusion matrix of the test data with the number of true positives, true negatives, false positives, and false negatives.

**Figure 4 diagnostics-15-01796-f004:**
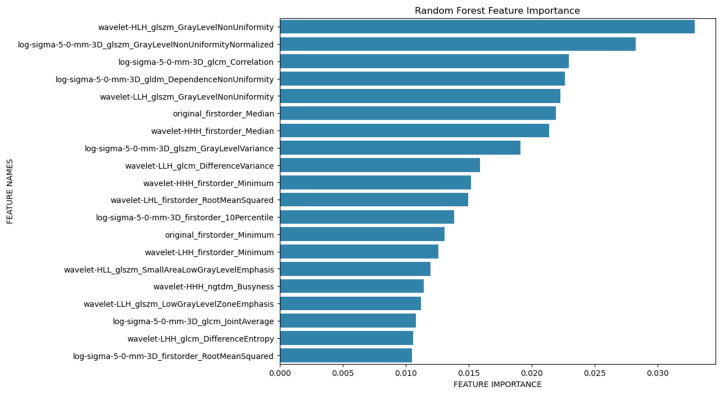
Feature importance plot.

**Figure 5 diagnostics-15-01796-f005:**
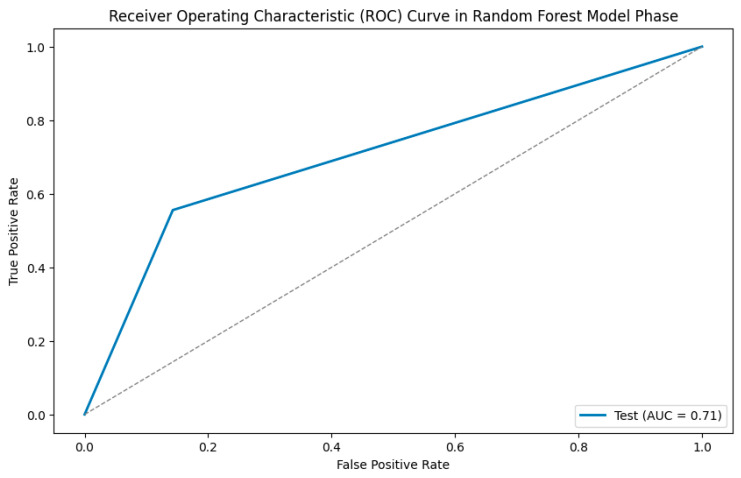
ROC analysis.

**Table 1 diagnostics-15-01796-t001:** Patient overview.

	Overall	Age 50–60 Years	Age 70–80 Years	*p*-Value
Number	90	54	36	0.074
Sex	42 male	30 male	12 male	0.052
	48 female	24 female	24 female	
Age	63.6	56.5	74.2	<0.001
	(51–79)	(51–60)	(70–79)	
Agatston Score	28.9	29.1	28.5	0.921
	(0–99.2)	(0–98.3)	(0–99.2)	
Hypertonia	22/42	12/24	10/18	0.764
	(52.4%)	(50.0%)	(55.6%)	
Hypercholesterinemia	22/42	14/24	8/18	0.537
	(52.4%)	(58.3%)	(44.4%)	
Diabetes mellitus	6/42	1/24	5/18	0.068
	(14.3%)	(4.2%)	(27.8%)	
Nicotine abuse	12/42	8/24	4/18	0.506
	(28.6%)	(33.3%)	(22.2%)	

**Table 2 diagnostics-15-01796-t002:** Overview of epicardial adipose tissue density and thickness. Abbreviations: HU: Hounsfield’s units, SD: standard deviation.

	Overall (Mean ± SD; Median (IQR))	Age 50–60 Years (Mean ± SD; Median (IQR))	Age 70–80 Years (Mean ± SD; Median (IQR))	*p*-Value
Number	90	54	36	0.074
Epicardial Fat Mean (HU)	−105.18 ± 20.95; −109.21 (29.25)	−104.99 ± 21.55; −108.62 (23.48)	−105.48 ± 20.03; −110.01 (33.26)	0.914
Epicardial Fat SD (HU)	38.81 ± 10.74; 36.54 (14.17)	39.96 ± 11.86; 36.34 (14.58)	37.1 ± 8.52; 38.75 (11.10)	0.454
Epicardial Fat Min (HU)	−199.90 ± 32.34; −195 (36.25)	−199.72 ± 33.94; −191.5 (31)	−200.17 ± 29.76; −204.5 (40.50)	0.561
Epicardial Fat Max (HU)	2.57 ± 40.96; −5 (54)	6.87 ± 42.73; −3.5 (57.25)	−3.89 ± 37.23; −9 (50.75)	0.227
Mean Diameter of epicardial adipose tissue (mm) Position 1	0.96 ± 0.38; 0.89 (0.41)	0.9 ± 0.38; 0.87 (0.43)	1.04 ± 0.37; 0.95 (0.4)	0.231
Mean Diameter of epicardial adipose tissue (mm) Position 2	0.7 ± 0.22; 0.66 (0.32)	0.7 ± 0.22; 0.63 (0.25)	0.71 ± 0.23; 0.72 (0.36)	0.613
Mean Diameter of epicardial adipose tissue (mm) Position 3	0.73 ± 0.25; 0.73 (0.35	0.74 ± 0.25; 0.74 (0.34)	0.71 ± 0.24; 0.71 (0.38)	0.664

**Table 3 diagnostics-15-01796-t003:** Results of the random forest (RF) classifier for predicting the age group based on left ventricular myocardial radiomics features.

**Train Data**				
	**Precision**	**Recall**	**F1-Score**	**Support**
Age Group 1	1	0.92	0.96	38
Age Group 2	0.89	1	0.94	25
accuracy			0.95	63
macro avg	0.95	0.96	0.95	63
weighted avg	0.96	0.95	0.95	63
**Test Data**				
	**Precision**	**Recall**	**F1-Score**	**Support**
Age Group 1	0.74	0.88	0.8	16
Age Group 2	0.75	0.55	0.63	11
accuracy			0.74	27
macro avg	0.74	0.71	0.72	27
weighted avg	0.74	0.74	0.73	27

## Data Availability

The data presented in this study are available on request from the corresponding author due to privacy concerns and ongoing research use.

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
