# Peer review of "Radiomics Signature of Aging Myocardium in Cardiac Photon-Counting Computed Tomography"

_diagnostics, 2025, doi:10.3390/diagnostics15141796_

Round 1
Reviewer 1 Report
Comments and Suggestions for Authors
I present my review in subsections, which should be addressed in the response to the reviewer and, if possible, improve and enhance the text of the article.
1. Although the Agatston score was adjusted, the text itself acknowledges the existence of differences in diabetes prevalence (p=0.068). How do the authors assess the possibility that the observed differences in texture are related to diabetes or other risk factors, rather than the ageing process itself? Could multivariate analysis be included or diabetic patients excluded to verify these effects?
2. The classifier results (accuracy ~0.74) are based on a relatively small test sample. How do the authors assess the risk of overfitting and the limited generalisability? Was cross-validation applied at the level of the entire dataset, for example? Please consider providing additional metrics (AUC, F1-score) and a more complete validation.
3. Segmentation was semi-automatic and performed by a single radiologist with 4 years of experience. Was inter-observer or test-retest repeatability assessment considered? How sensitive might the results be to subjective segmentation decisions?
4. The article suggests a potential biomarker of myocardial ageing, but without reference to threshold values or their clinical interpretation. Can the authors provide more practical recommendations or strategies for clinical validation in future studies?
5. The introduction lacks information introducing the reader to AI in medicine. Please read and and cite the following article DOI: 10.3390/diagnostics13152582.
Author Response
We sincerely thank all reviewers for their thoughtful and constructive comments, which have significantly helped us to improve the clarity, scientific quality, and overall value of our manuscript.
Reviewer 1
I present my review in subsections, which should be addressed in the response to the reviewer and, if possible, improve and enhance the text of the article.
Although the Agatston score was adjusted, the text itself acknowledges the existence of differences in diabetes prevalence (p=0.068). How do the authors assess the possibility that the observed differences in texture are related to diabetes or other risk factors, rather than the ageing process itself? Could multivariate analysis be included or diabetic patients excluded to verify these effects?
We thank the reviewer for pointing out the potential confounding effect of diabetes. While diabetes could indeed influence myocardial texture, we believe that its effect is likely minimal in our study. Although the difference in diabetes prevalence between the two age groups was noted (p = 0.068), this did not reach statistical significance. In contrast, the difference in age between the two groups was highly significant (p < 0.001), indicating that the observed texture differences are far more strongly associated with age rather than diabetes.
Given the clear statistical significance of age and the relatively weak association with diabetes, we conclude that age is the primary factor contributing to the observed myocardial texture changes. These findings suggest that the differences in texture are more likely driven by the ageing process itself.
Moreover, due to the retrospective nature of our study, information on comorbidities such as diabetes was not fully available for all participants, limiting the reliability of any extended statistical modeling. Since the primary focus of this study was to explore age-related myocardial changes, and considering the exploratory character and sample size, we did not perform a multivariate analysis. However, we acknowledge this as a limitation and agree that future studies with larger, prospectively collected cohorts and more complete clinical data should further investigate the influence of additional risk factors such as diabetes.
The classifier results (accuracy ~0.74) are based on a relatively small test sample. How do the authors assess the risk of overfitting and the limited generalisability? Was cross-validation applied at the level of the entire dataset, for example? Please consider providing additional metrics (AUC, F1-score) and a more complete validation.
We thank the reviewer for this important comment. To address the potential risk of overfitting and limited generalisability due to the relatively small test sample, we applied cross-validation at the level of the entire dataset. This helped ensure that the feature selection and model training were not overly tailored to a specific subset of data.
During our evaluation, we systematically tested different train/test split ratios. A 75/25 split was found to be optimal in terms of test accuracy. A 70/30 split led to substantially lower performance, likely due to an underpowered training set, while an 80/20 split showed signs of overfitting, with artificially inflated training accuracy but poorer generalisation to the test set.
In addition to reporting accuracy, we now provide a ROC curve analysis, which yielded an area under the curve (AUC) of 0.71 on the test data, indicating moderate discriminative performance. The corresponding ROC curve has been added as a new figure in the revised manuscript. These additions provide a more complete picture of model performance and help to further substantiate the robustness of our findings. Corresponding passages have been added in methods and results.
- Segmentation was semi-automatic and performed by a single radiologist with 4 years of experience. Was inter-observer or test-retest repeatability assessment considered? How sensitive might the results be to subjective segmentation decisions?
Thank you. A second radiologist controlled the segmentations. We added this to the paper: „To ensure the quality and consistency of the segmentations, all results were reviewed in a structured manner by a second radiologist with over a decade of experience in cardiovascular imaging and more than eight years of expertise in image segmentation.“
- The article suggests a potential biomarker of myocardial ageing, but without reference to threshold values or their clinical interpretation. Can the authors provide more practical recommendations or strategies for clinical validation in future studies?
We thank the reviewer for this valuable comment. In response, we have added additional analyses and explanations regarding the distribution and potential interpretation of the key radiomics feature, Wavelet-HLH_glszm_GrayLevelNonUniformity. Specifically, we now report in the Results section that no patient in the older age group exceeded a value of 1921, and no patient in the younger group fell below 729, suggesting that extremely high or low values may hold clinical relevance.
While we acknowledge that the definition of a strict diagnostic threshold is not feasible based on the current dataset, we have included a corresponding statement in the Discussion, proposing that values outside this observed range could potentially serve as early indicators of myocardial ageing. Moreover, we emphasize that future clinical validation studies should aim to confirm these findings, ideally in larger prospective cohorts, and assess their association with functional parameters such as strain or diastolic dysfunction.
These additions have been included in the revised manuscript (Results and Discussion sections) to provide more practical perspectives and strategies for future clinical implementation.
The introduction lacks information introducing the reader to AI in medicine. Please read and and cite the following article DOI: 10.3390/diagnostics13152582.
Thank you for this comment. We added this to the introduction: „In this context, the integration of artificial intelligence (AI) into clinical routine is of in-creasing interest, summarizing machine learning, artificial neural networks, and deep learning techniques to enhance diagnostic and treatment processes.“
Reviewer 2 Report
Comments and Suggestions for Authors
The article aims to analyze the changes seen in myocardial tissues with aging with radiomic features using Photon Counting CT (PCCT). Although the study is generally well designed, some improvements are needed.
1-A paragraph that better explains the motivation of the study should be added to the introduction.
2- It was observed that the patients included two age groups. However, why these age ranges were chosen should be explained. Why was the 60-70 age range excluded? Could this be due to misclassification of older individuals?
3-Explanatory information about dataset splitting (train/test) and parameter optimization should be provided.
4- ROC eğrisi, AUC değeri gibi ilave metrikler eklenmesi faydalı olacaktır.
5-In the discussion section, it is suggested that PCCT be examined in terms of cost, accessibility, and suitability for routine use.
Author Response
We sincerely thank all reviewers for their thoughtful and constructive comments, which have significantly helped us to improve the clarity, scientific quality, and overall value of our manuscript.
Reviewer 2
The article aims to analyze the changes seen in myocardial tissues with aging with radiomic features using Photon Counting CT (PCCT). Although the study is generally well designed, some improvements are needed.
1-A paragraph that better explains the motivation of the study should be added to the introduction.
Thank you for your comment. We added a motivation paragraph to the introduction: „As longevity is of rising interest in the medical community and the focus is lead from an aging society to a longevity society, there is an urgent need to better understand the parameters of myocardial aging. Potential myocardial changes during aging might become an identifier for unhealthy aging and a potential target for long-lasting longevity therapies. Hence, this study aims to explore potential age-dependent texture changes of the left-ventricular myocardium as well as potential differences in EAT density and thickness to identify novel cardiac biomarkers.“
2- It was observed that the patients included two age groups. However, why these age ranges were chosen should be explained. Why was the 60-70 age range excluded? Could this be due to misclassification of older individuals?
Thank you. We clarified this accordingly: „The age categories were defined using 10-year intervals to avoid borderline cases and ensure clear and unambiguous classification into distinct age groups, thereby facilitating more accurate analysis of aging-related factors.“
3-Explanatory information about dataset splitting (train/test) and parameter optimization should be provided.
We thank the reviewer for this helpful suggestion. We have now added further details on dataset splitting and model training to the revised Methods section. Specifically, the dataset was randomly split into 75% training and 25% test data, using a stratified sampling approach to ensure that both subsets reflected the same proportion of the two age groups (60% vs. 40%) as in the overall dataset. This approach was chosen to preserve class balance and ensure robust performance evaluation on the independent test set.
4- It would be useful to add additional metrics such as ROC curve, AUC value.
We thank the reviewer for this suggestion. As requested, we have now added a ROC curve analysis, which yielded an AUC of 0.71 on the test dataset. The corresponding ROC curve is presented in Figure 5, and a summary of the results has been included in the revised Results section.
5-In the discussion section, it is suggested that PCCT be examined in terms of cost, accessibility, and suitability for routine use.
We addressed this accordingly: „The limited global availability of the recently implemented PCCT technology reflects broader challenges in healthcare infrastructure worldwide. Due to its current restricted accessibility, the integration of PCCT findings into routine clinical practice remains limited. Although initial progress has been made through the development of more cost-effective alternatives, widespread implementation is still a long-term goal. As a result, the benefits and insights gained from PCCT are, for the time being, largely confined to specialized centers.“
Reviewer 3 Report
Comments and Suggestions for Authors
In general, the paper clearly presents the working tools and paradigm, with clear details on the results obtained. However, there are still some aspects of formatting and writing that need to be improved. On the scientific side - they have no relevance confusion matrix on train data, only the test data results are important - the classification rate on test data should also be reported - comparison with other methods existing in the literature - we can talk about The feature importance analysis only when some of the features are kept and the classification is resumed only with these, otherwise there are only assumptions and an unfinished analysis, which is not after completion. Other observations - formatting on the template - Table 3 is in the picture and should be as a table with text
Author Response
We sincerely thank all reviewers for their thoughtful and constructive comments, which have significantly helped us to improve the clarity, scientific quality, and overall value of our manuscript.
Reviewer 3
In general, the paper clearly presents the working tools and paradigm, with clear details on the results obtained. However, there are still some aspects of formatting and writing that need to be improved. On the scientific side - they have no relevance confusion matrix on train data, only the test data results are important - the classification rate on test data should also be reported - comparison with other methods existing in the literature - we can talk about The feature importance analysis only when some of the features are kept and the classification is resumed only with these, otherwise there are only assumptions and an unfinished analysis, which is not after completion. Other observations - formatting on the template - Table 3 is in the picture and should be as a table with text
We thank the reviewer for these constructive and detailed suggestions. In response:
– We have removed the confusion matrix for the training data, as suggested, and now report only the test performance. The classification accuracy on the test set is now clearly stated in the revised Results section and supported by additional performance metrics (precision, recall, F1-score, AUC).
– As recommended, we conducted an additional experiment where a new random forest classifier was trained using only the top 50 features identified in the feature importance analysis. However, this model performed worse than the full-feature model, with a test accuracy of 0.70 compared to 0.74 previously. Please see the image of the results in the Word File "Response to Reviewer".
This result highlights that although certain features contribute more heavily to classification, the model relies on a complex, multidimensional signature, and excluding lower-ranked features leads to a loss of relevant information. We have included this finding in the Discussion to clarify the role of the top features.
– Furthermore, we corrected all formatting issues and replaced Table 3 (and others) with fully formatted Word tables according to the journal’s template requirements.

Round 2
Reviewer 1 Report
Comments and Suggestions for Authors
The authors followed the reviewer's advice and recommendation thereby improving the manuscript. It has also become more readable. He therefore recommends the article for further editorial process.
Reviewer 2 Report
Comments and Suggestions for Authors
The authors have fulfilled all requests made of them.